# Modified Bacteriophage for Tumor Detection and Targeted Therapy

**DOI:** 10.3390/nano13040665

**Published:** 2023-02-08

**Authors:** Yuanzhao Shen, Jingyu Wang, Yuting Li, Chih-Tsung Yang, Xin Zhou

**Affiliations:** 1College of Veterinary Medicine, Institute of Comparative Medicine, Yangzhou University, Yangzhou 225009, China; 2Jiangsu Co-Innovation Center for Prevention and Control of Important Animal Infectious Diseases and Zoonoses, Yangzhou University, Yangzhou 225009, China; 3Joint International Research Laboratory of Agriculture and Agri-Product Safety, The Ministry of Education of China, Yangzhou University, Yangzhou 225009, China; 4Future Industries Institute, Mawson Lakes Campus, University of South Australia, Adelaide, SA 5095, Australia

**Keywords:** bacteriophage (phage), genetically engineered phages, phage display, chemical modification, biological detection, tumor screening, targeted therapy

## Abstract

Malignant tumor is one of the leading causes of death in human beings. In recent years, bacteriophages (phages), a natural bacterial virus, have been genetically engineered for use as a probe for the detection of antigens that are highly expressed in tumor cells and as an anti-tumor reagent. Furthermore, phages can also be chemically modified and assembled with a variety of nanoparticles to form a new organic/inorganic composite, thus extending the application of phages in biological detection and tumor therapeutic. This review summarizes the studies on genetically engineered and chemically modified phages in the diagnosis and targeting therapy of tumors in recent years. We discuss the advantages and limitations of modified phages in practical applications and propose suitable application scenarios based on these modified phages.

## 1. Introduction

Due to their increasing incidence and high mortality, malignant tumors are one of the main culprits threatening the life and health of all mankind [1,2]. Hanahan et al. summarized six characteristics in the development of malignant tumors including sustaining proliferative signaling, evading growth suppressors, resisting cell death, enabling replicative immortality, inducing angiogenesis, activating invasion, and metastasis [3]. Chemotherapy is one of the main treatments for malignancies, but chemotherapy often results in severe side effects for patients and has little effect in treating malignancies that have spread to other organs [4,5,6]. Early detection can help to slow down the tumor progression and in turn improve the cure rate. If the tumor in situ can be detected before it migrates, the survival rate and the effectiveness of surgical treatment can be significantly improved. Therefore, there is a pressing need to develop ultra-sensitive tumor diagnostic methods and effective therapeutic technologies.

Benefitting from the continuous exploration of nanotechnology, the application of nanomaterials in life sciences has become possible. Bacteriophages, as a natural nano-biomaterial, are gradually gaining attention due to the advantages of being able to be mass produced and their excellent modifiability, and have shown great potential in the fields of bio-related applications, materials science and soft matter, and physical chemistry [7]. Therefore, explorations in the use of phages for the diagnosis and treatment of cancer are now receiving increased attention. Phages have demonstrated great clinical potential in the diagnosis and treatment of cancer because phages can be selected for target cells, which is safer than traditional drug treatments, and phage therapy is less expensive than many other cancer therapies. Although phage therapy has made great progress in recent years, much effort is needed to understand and optimize this therapy for clinical use [8]. The development of gene recombination technology has greatly promoted the application of bacteriophage (phage) display technology in the diagnosis and therapy of malignant tumors [9]. Phages are a class of viruses with simple structure, containing only a single nucleic acid and capsid proteins. Because the displayed protein originates from the genetic sequence on the phage genome, phage display technology is widely used to screen protein sequences with the specific affinity to target molecules or to a variety of materials including organic, inorganic, and metallic materials. Icosahedral and filamentous phages are commonly used for phage modification, which includes gene modification and chemical modification [10,11]. In the review, the basic principles and methods of phage modification and the application of modified phages in tumor diagnosis and targeted therapy are introduced and discussed.

## 2. Phage Modification

The introduction of amino acids, peptides, or protein sequences into capsid genes to prepare recombinant phages is called gene modification or phage display [12]. Modified phages aim to express exogenous proteins that target specific sites (e.g., tumor antigens), and with this property, genetically modified phages can act as drug carriers to deliver drugs to specific sites. The most commonly used phages for modification are those with icosahedral heads such as T7, T4, lambda, and filamentous phages such as the M13 phage [11,13]. Because of the better understanding of their genetic information, these phages can be modified for diagnosis and targeted therapy. On the other hand, chemical modification aims to modify specific functional groups (amino, carboxyl, sulfhydryl, and phenol) on the surface of the phage by chemical coupling reactions. Some of these functional groups are derived from the phage itself, while others are genetically modified to display on the phage surface (Figure 1) [14]. In general, genetic modification enables the recombinant phage to have targeting specificity, while chemical modification can further endow a genetically modified phage with functional molecules or nanoparticles to obtain distinctive properties such as luminescence, endothermic, or detectable under physical conditions.

### 2.1. Genetic Modification of Phage

The genetic modification of phages was developed from the technology of homologous recombination. Homologous recombination technology was developed in the 1960s using phages and/or natural recombination systems to achieve efficient targeted DNA integration for gene knockout and gene knock-in [15]. It was complex to perform homologous recombination techniques until the 1980s. It was found that the flanking of the phage could be modified when researchers discovered that two different recombinant enzymes recognized flanks that caused deletions, insertions, or reversals on *E. coli* chromosomes in a site-specific manner [16]. However, there are some disadvantages of this method including the low recombination rate, difficulty in subsequent phage screening, and laborious process.

Another potential problem is that in the vitro genetic modification of phages is prone to mutations [17]. To this end, in 2008, Marinelli et al. invented the technology called ”Bacteriophage Recombineering of Electroporated DNA ”(BRED) [18]. The BRED aims to simultaneously transmit phage DNA and target DNA fragments into the host through electroporation to construct marker-free deletions, internal deletions, point mutations, and meaningless mutations of essential and non-essential genes. This approach allows the induction of targeted mutations at designated sites in chimeric phages, with higher productivity and better phage screening. On the other hand, the electroporation can reduce the mutation rate and improve the homologous recombination rate. By using BRED, Marinelli et al. successfully manipulated the genomes of phages such as Giles, TM4, Halo, BPs, and Che9c. To sum up, the BRED is a universal technique, and warrants attention in the efficiency of electro-transformation and the issue of the exogenous expression of plasmid.

Phage display technology (Figure 2) has been widely applied in the field of biomedicine. Based on the “donor DNA” structure, foreign sequences can be integrated into the phage genome by homologous recombination [19], and the recombinant phage can be further modified with amino acids, peptides, or foreign proteins. In other words, the phage display technology modifies the phage first, followed by the screening of the desired phage through multiple cycles. Phage display also contributes to the multivalent targeting method, that is, making the phage polyvalent by displaying different peptides at the head of each capsid. This method enhances the selective targeting effect for many times, thus improving the therapeutic efficiency and accuracy. For example, Haque et al. screened the phage-displayed library of peptides to identify peptides that selectively bind the kidney injury molecule-1 (KIM-1) and used this peptide to detect KIM-1-overexpressing tumors in vivo [20]. Ran et al. identified the unique sequences in the serum from patients with colon tumors, which provided theoretical support for clinical application using phage library technology [21]. Phage peptide display libraries can be used to find the affinity peptide for targeted molecules through various methods such as the peptides themselves, radiolabeled peptides, peptides bound to chemotherapeutic drugs, peptides fused with toxins, and nanoparticles carrying chemotherapeutic drugs. Wu et al. reported sensitive and selective bacterial detection using tetracystine (TC)-labeled phages combined with double arsenic dyes by cloning the DNA sequence of a tetracystine short peptide encoding 12 amino acids into the pIII protein gene of the M13KE phage. The benefit of this method is that, unlike luciferase and GFP, the smaller size makes the fusion of the TC marker with the pIII protein of M13KE less likely to disrupt its binding to the host cell surface receptors and the subsequent transmission of genetic material into the cell [22].

Phage modification can be performed easily thanks to the discovery of the new gen editing system CRISPR-Cas9. The CRISPR-Cas9 system can recognize and cleave foreign nucleic acids that invade bacteria, which is the mechanism of bacterial defense [23]. The CRISPR-Cas system integrates the target gene into the bacterial genome or encodes it into the plasmid. In 2014, Kiro et al. modified the T7 phage by homologous recombinant knockout and spliced it with the *E. coli* I-E CRISPR-Cas system to construct an efficient mutant [24]. Similarly, in 2017, Lemay et al. edited the gene of Lactobacillus phage P2 with CRISPR-Cas9. This study effectively addressed the difficulty of gene editing in P2 phages and demonstrated the reproducibility of the method, which promises to significantly expand the understanding of phage DNA host interactions [25]. The use of the CRISPR-Cas system to modify phages can yield a higher positive rate and reduce the time of phage screening, but the CRISPR-Cas system has not been widely used for the genetic modification of phages and warrants further development.

In recent years, with the rapid development of synthetic biology, in vitro expression systems or extracellular transcription translation systems have been widely applied in fields such as mRNA therapeutics and pathogen detection [26,27]. The above applications were first applied to protein synthesis until its prospects in the technical direction of phage recombination were discovered. Applications of synthetic biology in the field of phage include, but are not limited to, the use of amino acids, peptides, and decorative proteins on phage modification. By adjusting the amount or type of amino acids in the phage capsid protein, the density and class of active functional groups in the side chain can be affected [28]. Tridgett et al. expressed additional lysine residues on the outer surface of the pVIII protein of M13. This method enabled up to 520 additional exogenous groups to be attached to the surface of the phage by amine-directed conjugation. These results could help develop high-payload drug delivery nanocomponents and highly sensitive pathogen detection systems [29]. Peptides modify phages by expressing peptide motifs that can be recognized by enzymes into capsid proteins. Specific sites modified by enzymes are generated following the assembly of the capsid [30]. Oślizło et al. provided an affinity chromatography-based phage purification protocol using T4 phage as a template. Considering that the practice of the permanent introduction of foreign DNA into the phage genome could be a disadvantage for phage modification in medical applications, T4 phages have therefore been multiplied in fused bacteria expressing proteins with affinity tags from bacterial plasmids. The results show that this protocol is highly efficient and does not affect the purpose of preparing purified antimicrobial active phages for therapeutic use [31]. Shivachandra et al. developed an in vitro display system to express macromolecular protein (130 kDa) on the surface of the T4 phage capsid, which is a protective protein (PA)-highly antigenic outer capsid protein (hoc)-specific fusion protein. This technology was applied in the development of the Bacillus anthracis vaccine, which proves the feasibility of genetically engineered phages in the development of vaccines [32].

### 2.2. Chemical Modification of Phage

The surface proteins of the phage comprise a variety of amino acids, and the functional groups of amino acids make it possible to chemically modify phages with other molecules. At present, the chemical modification of phages mainly includes the amino group, phosphate group, phenol group, aldehyde group, and non-natural amino acids. The pathways of the chemical modification of phages are shown in Figure 3. These functional groups are inserted into specific molecules (phenol, Figure 3A; amino, carboxyl, Figure 3B; sulfhydryl, Figure 3C; phenol, unnatural amino, Figure 3D, etc.) on the surface of the phage by coupling reactions. Some of these active groups come from the phage itself or are inserted into the phage surface through genetic modification [14]. In general, chemical modifications are made at the N-terminus of proteins or on lysine side chains. The pH value can drive the reaction toward the N-terminal α-amino group (pKa value ~8) or the α-amino group of lysine (pKa value ~10). The disadvantage of this method is the low specificity and potential side reactions [33], causing mixed modifiers [34]. To improve the specificity, the phage is chemically modified with suitable functional groups to reduce the infectivity of the phage. For example, according to the process of protein glycosylation, the phage was modified by adding azido sugar to the host cell, and the phage was simply coupled and modified [35]. Huang et al. developed an envelope labeling strategy to combine the replication-intercalation labeling of viral nucleic acid [36,37].

The most commonly modification of the phage surface is through the amino group. For example, the surface of filamentous phages contains thousands of pVIII proteins, and each of them exposes a ε-amino group. Jin et al. established a general kinetic model of amine modification by coupling small molecules such as biotin and fluorescent dyes to the main capsid protein pVIII [38]. In addition to small molecules, amino acids and peptides can also be fused to the N-terminus of the phage pIII protein or the phage pVIII protein by phage display technology. Using phage display technology, Scott et. al. fused the peptide of the cysteine residue with the N-terminal of the pVIII protein [39]. Jencks modified ketones with amino functional groups to generate the fd phage with substituents such as the fd phage biosensor [40]. In 2022, Chen et al. invented a phage using chloroxime to couple cysteine and constructed a chemically modified phage library. It is fast and stable for the bis-chloroxime reagent to bind Cys-Cys in natural peptides and proteins [41].

Experts for the amino modification of phage surface proteins, carboxyl groups at the C-terminal and side chains of aspartic acid and glutamic acid of phage capsid proteins can also be used for chemical modification. For example, an amino molecule can be attached to a carboxyl group based on this strategy. Bar et al. bound the drug hygromycin to the carboxyl group of the fUSE5-ZZ phage modified by a specific antibody through a covalent amide bond. The drug-loaded phage targeted the receptor on the cancer cell membrane through the specific antibody for endocytosis, intracellular degradation, and drug release, resulting in the growth inhibition of target cells in vitro, with a potentiation factor of >1000 compared with the corresponding free drug [42]. Niu et al. used the M13 phage to prepare water-soluble conductive polyaniline (PANi)/M13 composite nanoparticles [43] as a promising candidate material for the development of nano-electronic devices, sensors, and energy storage devices. To better control the final composite fiber morphology, it is necessary to enhance the electronegativity on the surface of the M13 phage. This study used excess glutaric anhydrous acid to couple to lysine groups on the surface of M13 phages to derive electronegative carboxyl groups. To study the potential application of the filamentous phage Fd for the development of bio-templated nanowires or biosensors, Korkmaz et al. expressed a large number of MMM (three methionine) groups on the surface of wild-type Fd phages through gene editing technology. Ultimately, MMM-Fd phages can bind gold nanoparticles more efficiently compared with wild-type Fd phages. The above-mentioned studies demonstrate the great potential of the chemical modification of filament phages for biosensing applications [44]. In addition, gold nanomaterial modified phages can be used not only for biosensing, but also for targeted therapy. For instance, Peng et al. developed a protocol for the modification of phages with gold nanorods for photothermal therapy through phage-specific capture of target bacteria [45].

Crosslinked molecules bearing aldehyde groups, for example, glutaraldehyde, are often used to modify phages to yield chemically and thermally stable crosslinked biomaterials [46]. Glutaraldehyde reacts with active functional groups of proteins including amines, mercaptans, phenols, and imidazoles, of which the ε-amino group of lysine reacts best with glutaraldehyde [47]. Kitov et al. coupled aldehydes with 2-aminobenzamide oxime (ABAO) derivatives by the M13 phage display technique within 1 h. The change in fluorescence generated by the modified M13 phage can serve as a platform for the development of new bioconjugation strategies, fluorogenic probes, and post-translational diversification of genetically-encoded libraries, which can also be further developed in biosensing and other fields [48].

Unnatural amino acids can also be inserted into the phage capsid protein to exert its function. Unnatural amino acid modification is defined as an amino acid that is not encoded by the organism’s native genetic code, which can be incorporated into phage capsid protein constructions to display unique functional groups, in turn leading to selective chemical production on the modified site. For example, selenium cysteine is an unnatural amino acid cysteine analogue that is less affected by the phage’s natural coat protein residues than the lysine and cysteine present on the surface of the M13 phages, and therefore binds better to small molecule reagents [49]. In a recent study, Chen et al. developed a phage display method for the rapid unbiased screening of covalent inhibitors [50]. This approach uses active linkers to form cyclic peptides on the phage surface, and simultaneously introduces an electrophilic “warhead” to covalently react with the nucleophiles on the target, successfully avoiding the problem of difficult selectivity control of the electrophiles used for covalent inhibitor screening based on traditional methods. Similarly, Zheng et al. reported a chemically modified phage library with 2-acetylphenylboronic acid (APBA) as a warhead to bind lysine to form a reversible covalent inhibitor for tumor treatment by reversible iminoborate formation, which is highly effective and sensitive compared with the phage display library [51]. What differs from the phage library described by Bogyo is the reversibility of the APBA warheads and the fact that covalent binding is no longer limited to catalytic residues.

Paminophenylalanine, an unnatural amino acid derivative from tyrosine, can react with the phage capsid through oxidative coupling mediated by sodium periodate. This method has been used to modify MS2 phage-like particles and shows selectivity, even in the presence of tyrosine. In one study, the oxidative coupling of amino phenylalanine (a synthetic amino derivative of tyrosine) to MS2 phage-like particles was mediated by sodium periodate, even in tyrosine selectivity [52].

Polyethene glycol (PEG) modification enables the internal stability of biologically active proteins by combining them with water-soluble polymers [53]. The inherent hydrophilic characteristic of the hydroxyl group of PEG significantly contributes to the good hydrophilicity of PEG-modified phages [54]. The PEGylation of proteins or particles extends their half-life in vivo by forming a steric hindrance to prevent the enzymatic digestion of proteins or particles [55,56]. Phage display allows for the separation of phage–antibody–antigen complexes through the selection of antibodies. It is also possible to use non-ionic polyethene glycol linkers to improve the solubility and biocompatibility of M13, fd, and G1 phage-modified NHS ester conjugates. This method relies on the covalent attachment (conjugation) of PEG molecules [57] to the primary amino groups of proteins [58]. In 2003, Yamamoto et al. prepared a phage library to express mutant TNF-α and derived a biologically active, lysine-deficient mutant TNF-α (mTNF-α-Lys(-)). This TNF-α had superior molecular homogeneity, exhibiting enhanced biological activity in vitro and anti-tumor therapeutic efficacy [59]. Kim et al. demonstrated that PEGylated virus-induced substantial levels of T-helper type 1-related cytokine release (IFN-g and IL-6) in neonates and immunized mice, respectively. It was the first study to demonstrate that PEGylation could increase the survival of infectious phages by delaying the immune response [60]. This finding suggests that PEGylated phages may improve the effectiveness of phage therapy.

## 3. Modified Phage for Tumor Diagnosis

Due to their high target-specificity, nanoprobes are very sensitive for pathogen and tumor diagnosis. The modified phage and nanomaterials can be assembled into nanocomposite structures as nanoprobes. At present, there are two universal ways to construct phage nanoprobes. One is to use a phage to assemble nanoparticles with antibodies to form a phage nanoprobe to target the tumor. The other is to modify the phage with antibodies or proteins by genetic engineering technology to form a recombinant phage nanoprobe. Furthermore, there are four methods for tumors diagnosis with modified phages: (1) Using the light scattering characteristics of the nanoparticle modified phage; (2) imaging agent modified phages; (3) phage mediated signal amplification of the tumor cell by PCR; (4) phage mediated formation of visual monoclonal plaques. According to the above classification, we summarize and discuss some research methods of modified phages for tumor diagnosis, and plot in table (Table 1) according to the characteristics of modifications.

### 3.1. Tumor Diagnosis with Nanoparticles Modified Phage

Nanoparticle modified phages have been used in fields such as biosensing [61], heavy metal detection [62,63], etc., and they can also be used for the visual detection of tumor cells. The modified phage can be viewed as an antibody-like nanoprobe that can recognize specific antigens on tumor cells. This method only requires a portable dark field microscope and a magnetic nanoparticle (MNP) probe. It is very convenient and cost-effective to use this method to detect tumor cells [64]. In addition, due to their strong localized surface plasmon resonance (LSPR) properties, simultaneous magnetic microparticles (MMP), gold nanoparticles (GNP), and other functionalized nanomaterials can scatter brightly colored light under a darkfield microscope, enabling visual counting of tumor cells [65]. Xiao et al. reported that phages can be functionalized with highly stable and uniformly dispersed AuNPs (40 nm) and Au/Ag/Au composite nanoprobes (33 nm) to form a sandwich complex, and the multicolor scattered light can be directly photographed by a high resolution darkfield microscope system for the detection of tumors [66]. 

### 3.2. Visual Diagnosis of Tumor Cell with Molecular Imaging Agent Modified Phage

Molecular imaging technology has been applied for the detection of tumors, and molecular imaging agent modified phages allow for the visual detection of tumors at the multi-cellular level. Phage display library is an excellent tool for identifying tumor-specific targeting peptides [67]. Using a phage peptide library, Yang et al. discovered that a TNFR closed-loop peptide assembled with PEG-PLGA nanocarriers could effectively inhibit TNF-α activation and the associated immune response [68]. Closed-loop peptides can be considerably prolonged in their half-life. It has been demonstrated that the bioengineered M13 phages displaying a closed-loop peptide can serve as fluorescent probes for the imaging of human KB tumor cells after dual modification of fluorescent dye and a cell-targeting motif [69]. Tissue biopsy is a widely used method for identifying tumors, but it remains invasive and risky for patients. In order to avoid the harm caused by tissue biopsy, Maisano et al. developed a novel tumor diagnosis method by detecting tumor exosomes [70]. In order to prove the applicability of this technique in the absence of clear exosome markers, this study used the constrained random phage display library (CX7C) to perform multi-cycle phage screening in wild-type isogeneic tumor-bearing mice induced by the tumor cells of Eμ-myc tumor-bearing mice, and finally obtained a phage clone (Φ8) with a high affinity for the target tumor, and chemically modified the FITC on the surface of phage Φ8, so that the method has the characteristics of visualization. Cao et al. used the bevacizumab-sensitive LS174T colorectal cancer model and phage 12-peptide display library to identify bevacizumab reactive peptide (BRP) for the development of a new class of phage molecular imaging probes [71]. The results showed that the probe was structurally stable, had low cytotoxicity, specifically bonded to bevacizumab-treated tumors, and could be directly observed with near-infrared dye IRdye800 staining. Lomakin et al. believe that phage display technology utilizing polypeptides fused by chimeric phages exposed to phage proteins is one of the most powerful tools to achieve the identification and selection of cell subsets and the peptide ligand determination of various receptors [72]. However, the elution and reuse of chimeric phages have always been problematic. To overcome these issues, Lomakin et al. developed a system where all assembled M13K07 phages carried the chimeric p3 protein and were combined with fluorescence-activated cell sorting (FACS) for the detection and identification of surface-exposed receptors on living cells. By modifying the peptide targeting the specific protein and the matching fluorescent protein on the pIII protein of the M13K07 phage by gene editing technology, the identification and quantification of target proteins and cells can be fulfilled by FACS technology. In addition, Ishina et al. designed phage vectors based on the fADL-1e phage vector. The fADL-1e-based phage vector system designed by Ishina et al. can search for receptor ligands to detect the tumor cells based on the specific targeting using leukocyte antigen II and B cell receptors [73].

### 3.3. Tumor Diagnosis with Phage-Based Immuno-PCR

Immuno-PCR technology provides the highly sensitive detection of specific biomarkers; however, this approach requires the labelling of the detection antibody with synthesized oligonucleotides, followed by polymerase chain reaction (PCR) [74,75] to amplify the signal of antigen-binding events [76]. The highly heterogeneous population of antibody–DNA conjugates leads to the intrinsic disadvantage of immune-PCR including the reduced affinity and specificity of the antibody for its antigen. To address this issue, Brasino et al. utilized gene modification of the phage to express an antibody-binding peptide containing photo-crosslinked non-canonical amino acids (pBPA), allowing them to be detected and identified simultaneously by real-time PCR amplification of phage genomes [77]. Phage anti-immune complex assay real-time fluorescence quantitative PCR was used to detect small molecules. The short peptide displayed by the M13 phage display technique specifically binds to the antibody complex to inhibit the non-competitive inhibition of small molecular substances to complete the successful detection. Compared with the single phage display technique, this method has higher sensitivity for tumor detection [78]. Recently, Hou et al. reported a flexible recombinant M13 phage in combination with a magnetic microparticle probe (MMP) that could be employed to detect circulating tumor cells (CTCs) at the single-cell level (Figure 4) [79]. The antibody functionalized MMP was used to capture the CTCs. The second probe is a genetically engineered M13 phage with an N-terminal end fused single-chain variable region fragment (scFv) that recognizes the carcinoembryonic antigen (CEA) overexpressed in tumor cells. A single CTC was captured by two probes to form a sandwich MMP/CTC complex, which can be separated by a magnetic field and amplified by PCR to achieve a single CTC detection in a serum sample. 

## 4. Modified Phage for Tumor Therapy

Tumor-targeted therapy is an effective method for treating tumors, and is also a focus of current tumor treatment [80]. Due to the maturity of phage modification technology, it is now convenient, safe, and efficient to modify novel molecules on phages for targeting tumor cells or tissues. This technology has been widely used in finding new tumor-targeting molecules and in the delivery of drugs through a targeted strategy [81]. Since the size of the phages is only nanometers long, they can cross biological barriers such as the low vascular fibrosis barrier [82]. Due to the high surface area to volume ratio, phages can effectively carry drugs for tumor treatment compared with large molecule chemotherapeutic agents [83]. The modified phages, as drug carriers, allow for specific targeting tumor cells while sparing normal cells [84]. By targeting tumor cells through phage display technology, the piggybacked drug enters the cells through endocytosis and it kills or inhibits the tumor cells. It also facilitates precise treatment with less side effects, avoiding high doses of drugs and reduces tumor recurrence [85]. Using modified phages for tumor treatment is a cost-effective and less time-consuming method. We will discuss the screening of novel ligands and short peptide drugs, the preparation of tumor-specific antibodies, and the delivery of chemotherapy drugs in the following section. According to the above classification, we summarize and discuss some research methods of modified phages for tumor therapy, and plot in table (Table 2) according to the characteristics of modifications.

### 4.1. Screening of Novel Ligands and Short Peptides as Tumor-Targeting Drugs

Currently, anti-tumor drugs are often low targeting, toxic, and prone to inducing side effects. Therefore, there is a pressing need to find high-efficient and low-toxic anti-tumor drugs. Artificial synthesized peptide drugs have the advantages of good stability, high activity, intense penetration, high affinity and specificity, and low toxicity. These features provide essential therapeutic value for tumor treatment. In contrast, many small molecular anti-tumor drugs have shortcomings of short circulation half-lives in vivo, poor water solubility, low bioavailability, unreasonable distribution in tissues, and side effects. For many years, these defects have limited the development of small molecular anti-tumor drugs. Phage gene-editing technology facilitates the discovery of potential peptide drugs by using a random phage peptide library to find the specific peptide to a target molecule. The heritability of gene-edited phages undoubtedly provides great convenience for the discovery of peptide drugs. However, considering the influence of the chemical composition and conformation, the screening of synthetic peptide drugs requires a large number of samples. In addition, it is difficult to construct an unbiased phage polypeptide library by random gene insertion. New technology is needed to assist in the construction of large capacity unbiased phage peptide libraries.

One bead-one compound (OBOC) refers to the method of synthesizing millions of random compounds so that each bead displays only one compound. It consists of three main steps: the preparation of compound libraries, the detection of library components, and the screening of the target [86]. Houghten et al. reported an efficient exponential compound synthesis method named split synthesis, which can be used to prepare combinatorial compound libraries on solid-phase supports [87]. This method divides the composite mixture into equal parts in advance. When the number of synthetic times is n for each reaction, if the type of compounds is k, the number of components in the composition library (N) is N = n^k^. Based on a split synthesis method for solid support [88], only one chemical entity can be displayed on each solid-phase support [89]. An OBOC library consisting of millions of vectors was screened for quality to incorporate D-amino acids, unnatural amino acids, and many other organic building blocks and secondary structures into the design of phage libraries. This method has been successfully utilized to discover peptides targeting tumor vascular endothelial cells or tumor cells [90]. In 2002, Liu et al. successfully applied the OBOC approach to discover ligands for unique cell surface receptors of prostate tumor, T- and B-cell lymphomas, ovarian and lung tumors [91]. Some of these peptides demonstrated a high specific imaging property in nude mice. Hao et al. used a highly pooled and encoded OBOC peptide library in conjunction with a high-strength screening approach to identify a peptide that binds the α4β1 integrin specifically (IC50 = 2 nM) and to activate lymphoma cells [92]. Tie-2 has the potential to be used as a target gene in the gene therapy of solid tumors [93]. Wu et al. provided short peptides that could specifically bind to cells that express Tie-2 using the phage peptide library. Using the 125I labelling method, it was demonstrated that GA5 is exclusively enriched in SPC-A1 tumor cells expressing Tie-2 and that the phage vector carrying the GA5 gene allows for greater transfection efficiency. When the vector containing the p53 gene and GA5 peptide was injected into the SPC-A1 tumor-bearing mice, the tumor volume was significantly reduced [94].

### 4.2. Validation of Specific Tumor-Targeting Antibodies through Phage Display Technology

Tumor antigens such as epidermal growth factor receptor (EGFR), human epidermal growth factor receptor 2 (HER2), and carcinoembryonic antigen (CEA) have significant diagnostic and prognostic values as tumor biomarkers. By modifying phages, a specific antibody library including antigen-binding fragments (Fab), single-chain antibodies (scFv), and other antibody fragments can be established. This technology is beneficial given its short cycle, easy operation, and affordable price. However, due to the small molecular antibodies generated from modified phage technology, it is difficult for them to stimulate the body’s immune response and allows them to reach the target cells from the intercellular space efficiently. To this end, the research focus has been to use modified phage technology to efficiently screen large molecular tumor-specific antibodies for targeted tumor therapy in recent years. In 2013, Ayat et al. constructed a pair of recombinant phage antibody libraries and successfully isolated human single-chain antibodies against the specific biomarkers of HER2 and CEA to target the lymph nodes of breast tumor patients [95]. In 2014, Lin et al. isolated the anti-human trophoblast cell surface antigen 2 (Trop2) Fab antibody from breast tumor cells using phage display technology and demonstrated that the anti-Trop2 antibody effectively reduced tumor migration and inhibited tumor growth. The anti-Trop2 antibody could also promote the death of breast tumor cells [96]. In addition, Han et al. obtained gene-edited phages specific to the total prostate-specific antigens (t-PSA) using the f8/8 landscape phage library after three rounds of biological screening. Furthermore, they also developed sandwich enzyme-linked immunosorbent assay (ELISA) and differential pulsed voltammetry (DPV) determination systems based on the phages, both of which achieved high specificity and reliability [97]. 

From 2014 to 2019, the U.S. Food and Drug Administration (FDA) approved three types of anti-tumor antibodies, all of which were discovered based on phage display techniques. Ramucirumab is an anti-tumor antibody that selectively binds to extracellular vascular endothelial growth factor receptor (VEGF)-2, thereby inhibiting the induction and angiogenesis of (VEGF)-2 and the proliferation and migration of endothelial cells. In 2014, Ramucirumab was approved by the FDA for the treatment of advanced gastric or gastroesophageal junction adenocarcinoma [98]. Similar to ramucirumab, necitumumab was developed using a Dyax Fab phage display library and was approved by the FDA in 2015 for use in conjunction with gemcitabine and cisplatin as a first-line treatment of metastatic non-small cell lung tumor (NSCLC) [99]. In 2017, the FDA approved avelumab, a fully human mAb, for the treatment of Merkel cell carcinoma patients over the age of 12. To sum up, it is promising to use phage display technology to find new anti-tumor antibodies.

### 4.3. Targeted Delivery Vehicle for Chemotherapy Drugs

Since the vast majority of phages are in the nanometer scale, they have the potential to penetrate biological barriers such as the capillaries [100]. The specific phages carrying drug molecules can achieve targeted tumor therapy and reduce the damage and side effects to normal cells [101]. Drug-loaded phages targeting tumor cells can enter the cell through endocytosis in which the drug inhibits or kills the tumor cell. Phage loading technology facilitates the precise treatment of tumors with less side effects, avoids drug overdose, and reduces tumor recurrence [80]. The modified phage treatment of tumors is a cost-effective and time-saving methodology. Phages serve as excellent carriers to deliver chemotherapeutic drugs to specific sites, and their targeting can be classified as passive targeting and active targeting. The nanoscale size of phages makes it challenging for them to be cleared by the endoplasmic reticulum system.

When the filamentous phage of class Ff infects the host bacteria, the capsid protein pVI is responsible for anchoring to the host cell membrane. Therefore, the pVI protein in vitro is lipophilic and can spontaneously bind to the lipid bilayer. By displaying polypeptide drugs on capsids or linking drug-carrying liposomes to specific polypeptides, the integrity of the polypeptide drugs would not be compromised [102], and the efficiency of the targeting of phages would also improve. A study by Pastorino et al. found that liposome-targeted therapy for gliomas using doxycycline liposomes can effectively inhibit tumor growth and is a valuable tool for glioma liposome-targeted therapy [103]. As shown in Figure 5, Choi et al. displayed a circular RGD (cRGD) peptide on the pVIII major capsid protein using recombinant DNA technology to construct an engineered phage to target tumor cells. The type 88 phage engineering method was used in this approach as the genome of the Type 88 bacteriophage has two distinct genes that encode the pVIII protein: a wild-type g8 and a recombinant g8. In this system, wild-type g8 can be used to help maintain the integrity of the phage pVIII protein, while recombinant g8 encodes a new master epidermal protein (recombinant pVIII) fused with a functional peptide/protein. In addition, recombinant pVIII can accommodate relatively large or structurally bulky inserts that the wild-type pVIII coat proteins cannot tolerate. Using the Type 88 phage engineering method, a new gene is inserted in the non-coding region of the phage genome to additionally express cRGD on the pVIII protein. Under the control of IPTG, we can adjust different amounts of cRGD peptides displayed on the phage particles up to 140 copies. Compared with the M13 phages modified with either linear RGD on pVIII or cRGD on pIII, cRGD on recombinant pVIII exhibited higher internalization efficiency in ligand density and conformational structure into HeLa cells [104]. By applying phage antibody libraries to disease proteomics studies, Nagano et al. developed a new drug delivery platform to validate many protein candidates [105]. Functional mutant proteins R1antTNF were generated with enhanced receptor affinity and specificity by using phage display technology to improve the in vivo stability of biologically active proteins.

Liposome encapsulated antitumor peptides, which were developed by phage display, are widely used in clinical tumor therapy. Hölig et al. reported a high-affinity cyclic RGD lipopeptide (RGD10) targeting αv-integrin and applied doxorubicin-loaded RGD10 liposomes in an in vivo C26 colon cancer mouse model. It showed an improved efficacy compared to free doxorubicin and non-targeted liposomes. RGD10 is a tripeptide sequence selected from the phage display library against endothelial cells and melanoma, and subsequently integrated into liposomes [106]. With the same principle, Wang et al. used an 8-mer landscape library f8/8 and a bio-panning protocol against MCF-7 cells to select a landscape phage protein bearing the MCF-7-specific peptide. Subsequently, a chimeric phage fusion coat protein specific toward MCF-7 cells, identified from a phage landscape library, was directly incorporated into the liposomal bilayer of doxorubicin-loaded PEGylated liposomes (Doxil) without additional conjugation with lipophilic moieties. The results show that phage-modified doxorubicin liposomes with PEGylated liposomes loaded with MCF-7 breast tumor cells demonstrated significantly increased cytotoxicity toward target cells in vitro [107]. In addition, Mudd et al. developed a binding protein-4-targeted bicyclic toxin adduct to synthesize the conjugate BT8009 for the treatment of solid tumors. The bicyclic peptide obtained by phage display can specifically binds to Nectin-4 of tumor cells [108].

## 5. Conclusions

Phage display technology has shown great value in screening peptides or antibodies that interact with the target molecules. Peptide libraries and antibody libraries based on phage display technology are used to screen phage clones, peptides, or antibodies with a high affinity to the target tumor cells. Due to the robust modification of various dyes on the proteins of phages, phage probes have been developed to visualize tumor cells. Furthermore, due to the advances in gene editing, chemical synthesis, and imaging techniques, modified phage probes provide new strategies for tumor diagnosis and treatment. Modified phages have the potential to revolutionize the diagnosis and prevention of diseases including but not limited to the treatment of tumors [109]. Tumor-targeted peptide screening identifies dysfunctional tumor blood vessels, tumor stromal cells, dense extracellular matrix, and overexpression receptors in solid tumors [83]. The modified phage can be used to synthesize multifunctional drugs to kill tumors. Various types of phages have undergone genetic modification and chemical modification for tumor diagnosis and treatment. Molecular imaging agents can be used for tumor detection and treatment, which shows that modified phages have great prospects in the field of medical diagnosis.

Although there are many promising reports on the use of modified phages in the diagnosis and treatment of tumors, there are still many challenges to overcome prior to their wide application. For example, chemically modified phages may be ineffective due to an immune response in the body. It has demonstrated that patients have an immune response to polyethene glycol modified drugs, which significantly impacts the clinical treatment effects [110]. The reasons and processes for this phenomenon have not been thoroughly investigated. In addition, it is also very difficult to ensure the specificity of the modified phage used for tumor diagnosis and treatment. In some cases, while a single-phase, uniform chain length of PEG does not enhance the bioavailability or extend half-life, it may still result in side products, even though the optimal conditions are adopted to enhance the reaction efficiency [111]. Capture efficiency is an essential determinant of the effectiveness of modified phages in detecting tumors. To enhance capture efficiency and successfully translate it into clinical practice, peptide targeting ligands must be optimized for their affinity and solubility in water. However, in animal experiments and clinical treatments, the number of modified phages targeting tumors following passing through blood circulation in the body would be considerably reduced. In practical applications, the effect of phages on tumor apoptosis is also not ideal. We need to develop technologies to maintain the phage targeting efficiency and apoptotic effect. There is still much room for the development of phage modification technology in clinical treatment. We envisage that much effort will be continuously devoted to the development of new modification technologies. It is believed that the modified phage technology will overcome many barriers and become a practical methodology in the clinical diagnosis and treatment of tumors in the near future.

## Figures and Tables

**Figure 1 nanomaterials-13-00665-f001:**
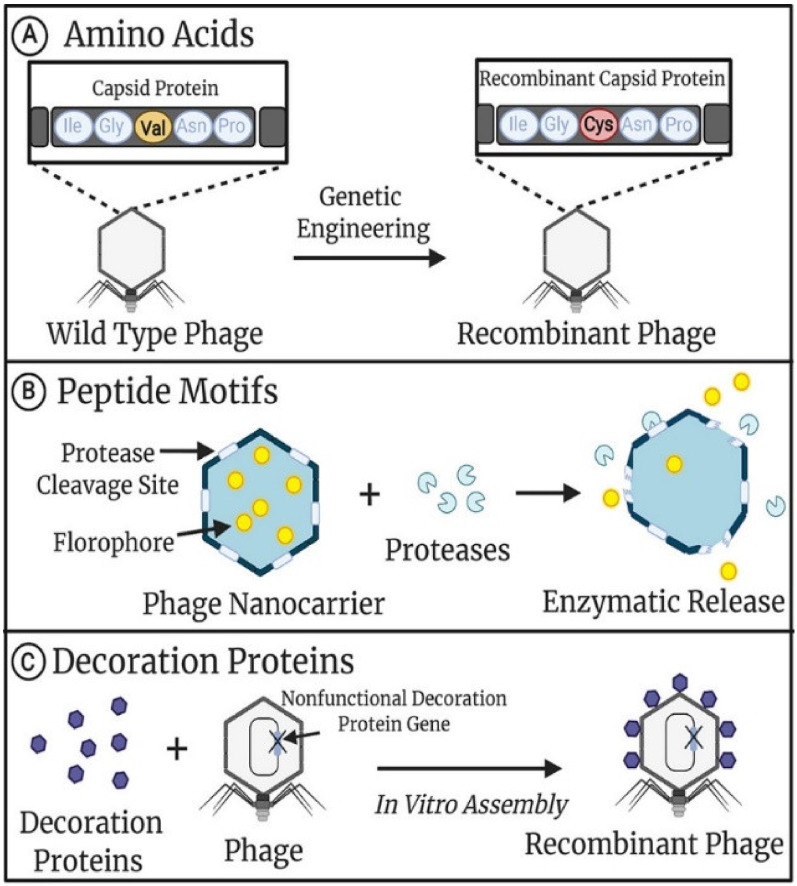
Genetic modifications of phage capsids. (**A**) Single amino acids in phage capsids can be substituted to alter the number and type of functional groups accessible for downstream chemical modification. (**B**) Peptide motifs recognized by specific enzymes can be incorporated into phage capsids for downstream enzymatic modification or the controlled release of contents. (**C**) Recombinant capsid decoration proteins can be synthesized separately from the phage and assembled to the capsid in vitro, allowing for large complex proteins to be displayed. This figure is reproduced from [14] (© 2021 American Chemical Society).

**Figure 2 nanomaterials-13-00665-f002:**
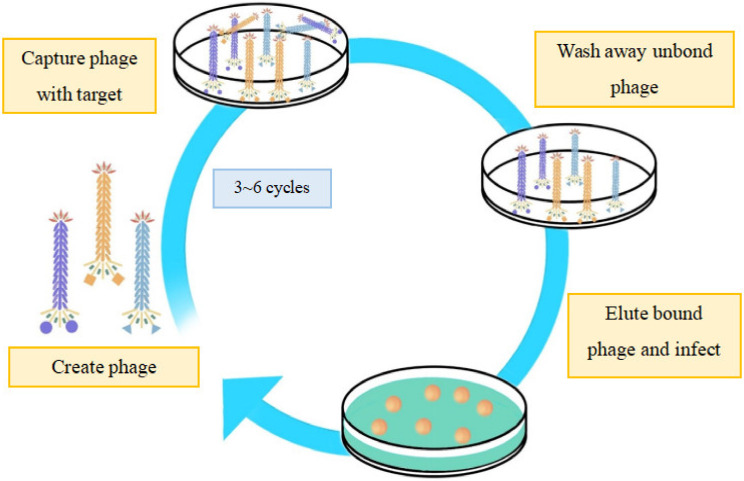
Illustration of the principal of phage display technology. The DNA sequence encoding foreign peptides or proteins is inserted into the gene encoding phage capsid protein, and the specific phage can be obtained after three to six cycles of “adsorption–elution–amplification”.

**Figure 3 nanomaterials-13-00665-f003:**
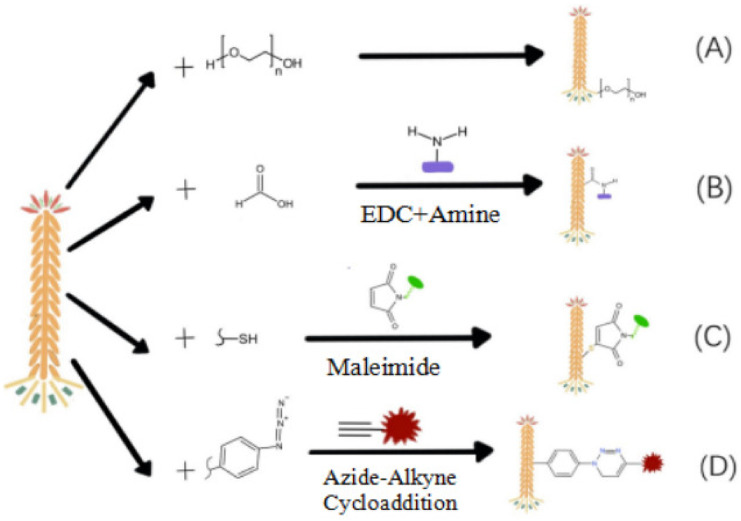
Schematic diagram of the pathways of the chemical modification of the phage. (**A**) Phage modified with phenol. (**B**) Phage modified with amine by EDC (1-(3-Dimethylaminopropyl)-3-ethylcarbodiimide hydrochloride). (**C**) Phage modified with sulfhydryl by Maleimide. (**D**) Phage modified with phenol and unnatural amino by Azide-Alkyne Cycloaddition.

**Figure 4 nanomaterials-13-00665-f004:**
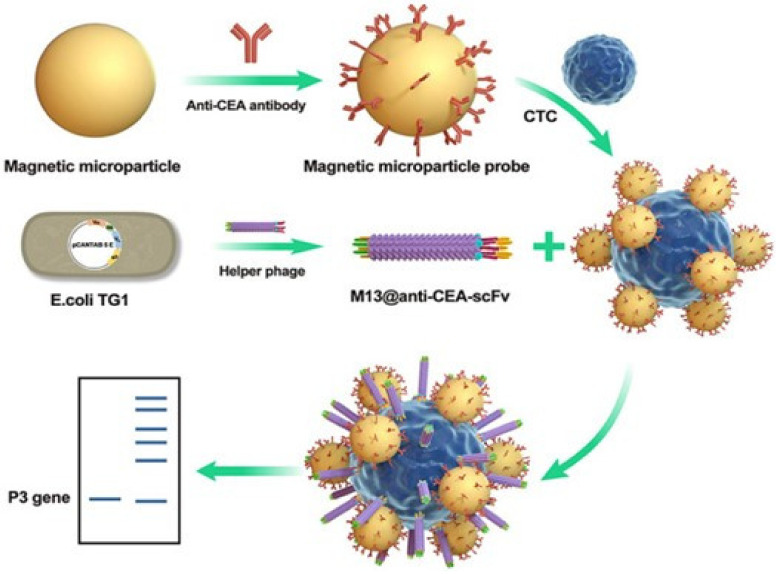
Schematic representation of an ultrasensitive CTC detection based on phage probes. This figure was reproduced from [79] (© 1969, Elsevier).

**Figure 5 nanomaterials-13-00665-f005:**
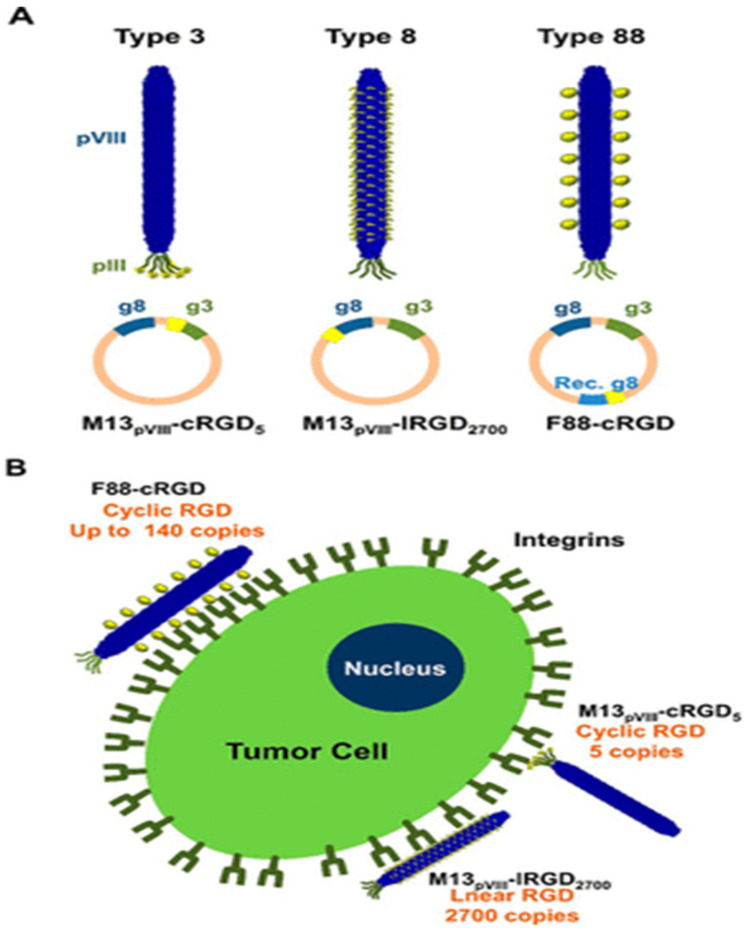
Schematic representation of three different types of phage engineering approaches and targeted delivery to cancer cells. (**A**) Incorporation of the phage with target peptides using Type 3, Type 8, and Type 88 approaches. The type 3 phage can display five copies of cyclic RGD on the pIII protein while the Type 8 phage can display 2700 copies of RGD, but only in linear form. Using Type 88 phage engineering, cyclic RGD can be incorporated in the recombinant pVIII protein up to 140 copies. (**B**) Schematic illustration of the targeted delivery of the engineered phage to cancer cells. This figure is reproduced from [104] (© 2013, American Chemical Society).

**Table 1 nanomaterials-13-00665-t001:** Summary of the cited reference of the modified phage for tumor diagnosis.

Reference No.	Species of Phage	Modified Site	Modified Material	Diagnosis Target
68	M13	P8 protein	Folic acid and fluorescent molecules	Human KB cancer cell
69	M13	P3 protein	Random peptides with a disulfide constrained loop	Exosome of Eμ-myc tumor
70	M13	P3 protein	Random 12-mer peptides	LS174T colorectal cancer cell
71	M13K07	Biotinylated P3 protein	PE-Cy7-Strept	Raji and Raji-FL cell
72	fADL-1e	3FLAG or hemagglutinin	B-cell receptor peptide ligand	Raji and Raji-FL cell
76	Fd derived filamentous phage	P3 protein with antibody binding peptide and pBPA	Mouse IgG1 clone Mab1 raised against recombinant TNFα/IL-6/IL-1β	TNFα/IL-6/IL-1β
78	M13	P3	Anti-CEA-scFv	Circulating tumor cell

**Table 2 nanomaterials-13-00665-t002:** Summary of the cited reference of the modified phages for tumor therapy.

Reference No.	Species of Phage	Modified Site	Modified Material	Target
93	M13	P3 protein	Peptide ligand and polyethylenimine	Tyrosine kinase with immunoglobulin and epidermal growth factor homology domain-2
94	M13	P3 protein	HER2 and CEA anti-scFv	HER2 and CEA antigen
95	M13	P3 protein	Anti-Trop2 Fab antibody	Human trophoblastic cell surface antigen 2
96	f8/8 phage	P8 protein	Fusion peptide	Total prostate-specific antigen
97	Dyax Fab phage display library	P3 protein	Anti-IgG1 Fab	Vascular endothelial growth factor receptor
98	Dyax Fab phage display library	P3 protein	Recombinant human IgG1 mAb	Metastatic non-small cell lung tumor

## Data Availability

Not applicable.

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
