# Peer review of "Modified Bacteriophage for Tumor Detection and Targeted Therapy"

_nanomaterials, 2023, doi:10.3390/nano13040665_

Round 1
Reviewer 1 Report
The authors provided a well-documented overview of the impact of phages on tumor diagnosis and therapy. In addition to this, this article provides an interesting starting point for a discussion on the next challenges for phage-passionated researchers. The field of research focused on the biomedical applications of bacteriophages is in continuous evolution and even if the article is well written the review could be improved by adding some recent works related to the possibility of specifically targeting and characterizing the tumor-derived exosomes in preclinical models ( see PMID: 35141731 and others). Good luck!
Author Response
1. The field of research focused on the biomedical applications of bacteriophages is in continuous evolution and even if the article is well written the review could be improved by adding some recent works related to the possibility of specifically targeting and characterizing the tumor-derived exosomes in preclinical models (see PMID: 35141731 and others).
Reply: Thanks for your valuable suggestion. The reference mentioned in your comments has been cited in the article (line 324).

Reviewer 2 Report
This review summarizes the studies on genetically engineered and chemically modified phage in diagnosis and targeting therapy of tumor in recent years.
Authors should add a summary table showing: the names or sequence peptides of the phage engineered, the chemical modifications, the tumor target cells and literature reference.
Author Response
1. Authors should add a summary table showing: the names or sequence peptides of the phage engineered, the chemical modifications, the tumor target cells and literature reference.
Reply: Thanks for your valuable suggestion. We have added a summary table containing the relevant references cited in “3. Modified phage for tumor diagnosis” and “4. Modified phage for tumor therapy” in the revised manuscript (please see line 384 and 555).

Reviewer 3 Report
I don’t think this review belongs in Nanomaterials. It hardly addresses the conjugation of phages with nanosystems and in the rare cases when it does, the discussion is very superficial. Submitting a contribution to the correct journal is important to target the correct readership. By submitting this review to Nanomaterials the authors appear to be oblivious to this key concept.
For this reason, I cannot recommend acceptance of this review.
Specifically, I find this contribution very superficial and, in some instances, wrong. The presentation also indicates a sloppy approach to preparing it.
1. The captions to Fig. 2 and 3 have been exchanged (Fig. 2 is not cited in the manuscript). The chemical modifications reported in Figure 2 (the caption is that of Fig. 3) are poorly described (case A is hardly understandable). Reference 13 is much better at representing (and discussing) these concepts.
2. Nanosystems are mentioned as a single line (line 221) and line 353 and as a small paragraph (2.1, ten lines). Is this sufficient to consider publication in a nanosystems journal?
3. Stating that phages can penetrate the blood-brain barrier because of their small size is simply wrong. As a matter of fact, even smaller drugs do not penetrate the blood-brain barrier! This statement is made twice.
As stated above I doubt a nanoscientist will find this review useful.
Author Response
1. The captions to Fig.2 and 3 have been exchanged( Fig.2 is not cited in the manuscript). The chemical modifications reported in Figure 2 (the caption is that of Fig. 3) are poorly described (case A ishardly understandable). Reference 13 is much better at representing (and discussing) these concepts.
Reply: Thanks for your valuable suggestion. We have corrected these. Fig.3 was drawn by ourselves and does not belong to any reference; At the same time, we revised the caption to make case A of Fig.2 easier to read.
2. Nanosystems are mentioned as a single line (line 221) and line 353 and as a small paragraph (2.1.ten lines). Is this sufficient to consider publication in a nanosystems journal?
Reply: Thanks for your comments. Bacteriophages are nanoscale, we consider them a standard biological nanomaterial, especially being assembled with metal nanomaterials. This review mainly discusses the application of phages in tumor diagnosis and therapy, which belongs to the scope of nanomaterial applications.
3. Stating that phages can penetrate the blood-brain barrier because of their small size is simply wrong. As a matter of fact, even smaller drugs do not penetrate the blood-brain barrier! This statement is made twice.
Reply: Thanks for your valuable suggestion. In the original text of the cited reference ([81]), it is not the phage that penetrates the blood-brain barrier, but the polypeptide expressed by the phage combined with the nanomaterial that penetrates the blood-brain barrier. We have corrected these two mistakes (please see line 392 and 491 in the revised manuscript).

Reviewer 4 Report
Yuanzhao Shen et al reported review of Modified Bacteriophage for Tumor Detection and Targeted Therapy. They discuss the advantages and limitations of modified phages in antibacterial and antitumor applications, depicting application developments based on modified phages.
As to me this review if written in good form, timely overviewing the survey and novelties in bacteriophage related targeted diagnostics and therapies.
However, one important aspect of phage-based therapies was not discussed: antibacterial photodynamic therapy (PDT). There are several lately published works dealing with this method like "Ran Bei et al (2021). A photo-sensitizable phage for multidrug-resistant Acinetobacter baumannii therapy and biofilm ablation". Chemical Science, 10.1039.D0SC04889E doi:10.1039/d0sc04889e".
In my opinion, adding antibacterial PDT chapter to the article would better describe the current use of bacteriophages and their modifications in medical treatments
Author Response
1. In my opinion, adding antibacterial PDT chapter to the article would better describe the current use of bacteriophages and their modifications in medical treatments.
Reply: Thanks for your suggestion. But we found the article you recommended describes a photodynamic phage therapy for multidrug-resistant bacteria. Our review focuses on tumor detection and targeted therapy. Therefore, we did not add the antibacterial PDT section in our manuscript.

Round 2
Reviewer 3 Report
The authors are correct in stating that the matter covered in this review fits within the journal's scope. Still, I am not particularly impressed by the quality of this contribution. I noticed that a similar one was published two years ago in this journal but has not been cited by the authors. This reference (10.3390/nano10101944) must be cited.
Author Response
1.The authors are correct in stating that the matter covered in this review fits within the journal's scope. Still, I am not particularly impressed by the quality of this contribution. I noticed that a similar one was published two years ago in this journal but has not been cited by the authors. This reference (10.3390/nano10101944) must be cited.
Reply:Thank for your valuable suggestion. The reference mentioned in your comments has been cited in the article (line 40).
Reviewer 4 Report
No comments
Author Response
1.No comments
Reply: Thank you for your valuable comments on this review.